# From Liquid to Solid: Cocrystallization as an Engineering Tool for the Solidification of Pyruvic Acid

Camila Caro Garrido [1,*] , Koen Robeyns [1] , Damien P. Debecker [1] , Patricia Luis [2] and Tom Leyssens [1,*]

1 Institute of Condensed Matter and Nanosciences, UCLouvain, 1 Place Louis Pasteur, B-1348 Louvain-la-Neuve, Belgium; koen.robeyns@uclouvain.be (K.R.); damien.debecker@uclouvain.be (D.P.D.)

2 Institute of Mechanics, Materials and Civil Engineering, UCLouvain, 2 Place Sainte Barbe, B-1348 Louvain-la-Neuve, Belgium; patricia.luis@uclouvain.be

* Correspondence: camila.carogarrido@uclouvain.be (C.C.G.); tom.leyssens@uclouvain.be (T.L.); Tel.: +32-0497881129 (C.C.G.); +32-10472811 (T.L.)

**Abstract:** Pyruvic acid is an organic compound used in various fields (e.g., the pharmaceutical, cosmetic, food, and chemical industries) and subject to constantly growing demand. Pyruvic acid is liquid at room temperature, rendering manipulation less straightforward. Furthermore, in the liquid phase, pyruvic acid is air-sensitive. We here present a multi-component crystal engineering strategy to render pyruvic acid solid under ambient conditions, focusing on cocrystallization and salt formation. Out of 73 screened cocrystal and salt formers, eight were found to form novel crystalline forms with pyruvic acid. Four of these were studied in detail, with pyruvic acid stable in a solid phase at temperatures up to 120 °C. These results illustrate the effectiveness of cocrystallization as a tool to convert unstable liquid compounds into stable crystalline solid forms.

**Keywords:** pyruvic acid; cocrystallization; crystal engineering; stabilization; solidification





## 1. Introduction

Compared to liquids, solids are often intrinsically more stable to temperature fluctuations, and show reduced air sensitivity, resulting in a longer shelf-life [1]. Furthermore, solids are easier to handle during transport and in manufacturing processes [2]. This is part of the reason why solids are often preferred in the food and pharmaceutical industries. The solid–liquid nature of a given compound is, however, not free of choice, as it depends on the intrinsic melting temperature of the compound.

Pyruvic acid (Figure 1), also known as 3-oxopropanoic or α-ketopropionic acid, shows a low melting temperature of 11.8 °C. Under standard conditions, pyruvic acid appears as an amber viscous liquid which is corrosive and air-sensitive. Indeed, pyruvic acid polymerizes via an aldol-like condensation reaction in aqueous solutions [3,4]. In addition to being involved in major energy metabolic pathways such as glycolysis, gluconeogenesis, the Krebs cycle, or fermentation [4], pyruvic acid is also a valuable starting material for the synthesis of various pharmaceuticals (such as L-alanine, L-tyrosine, L-tryptophan, L-DOPA, etc.), food additives, cosmetics, polymers, and crop protection agents. It is also used on its own as a flavouring agent to give a sour taste to foods, and for the treatment of acne [4,5].

Considering its wide applicability, it would be interesting to be able to transform pyruvic acid into a solid form. In this context, the inorganic salt formation of pyruvic acid has already been successfully applied, with calcium pyruvate used as a food supplement claimed to enhance physical endurance and to induce fat loss [6].

In this paper, we use a different approach to inorganic salt formation, focusing on multi-component crystal engineering, crystallizing pyruvic acid with organic compounds,

leading to cocrystals as well as salt and salt cocrystal. Cocrystallization [7,8] is a useful tool that has already been used to alter the melting point [9–13], solubility and dissolution rate [14–19], but also the stability [20,21], and mechanical properties of target compounds [22]. Moreover, it is an often-applied approach to control polymorphism, hygroscopicity, or deliquescence [23–26]. In this report, we use cocrystal engineering to obtain solid forms comprising pyruvic acid with improved physical properties.

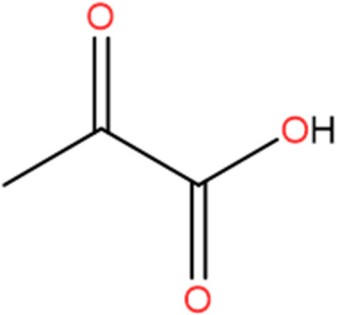

**Figure 1.** Chemical structure of pyruvic acid (oxygen atoms in red).

## 2. Materials and Methods

### 2.1. Materials

Pyruvic acid (CAS: 121-17-3; >97%), carbamazepine (CAS: 298-46-4; >97%), and hypoxanthine (CAS: 68-94-0; >98%) were purchased from TCI Europe N.V. (Zwijndrecht, Belgium), 4-nitrobenzamide (CAS: 619-80-7; 98%) and theophylline (CAS: 58-55-9; ≥99%) were acquired from MERCK (Hoeilaart, Belgium), adenine (CAS: 73-24-5; 99%) and isonicotinamide (CAS: 1453-82-3; 99%) were obtained from Thermo Fisher GmbH (Dreieich, Germany), and caffeine (CAS: 58-08-2; 98.5%) and nicotinamide (CAS: 98-92-0; 99%) were bought from Acros Organics (Belgium). Solvents were commercially available from VWR International BV (Brecht, Belgium). All the materials were used as received, without any further purification.

### 2.2. Solid Forms Screening by Grinding

A total of 73 cocrystal and salt formers typically used in cocrystallization and salt formation were selected among carboxylic acids, amides, amino acids, purines and pyrimidine (derivatives), and profens. The screening was performed through neat grinding with a MM 400 Mixer Mill grinder manufactured by RETSCH (Haan, Germany), equipped with two grinding jars, each able to contain five Eppendorf tubes of 2 mL. The tubes were filled with an equimolar amount (0.3 mmol) of pyruvic acid (PA) and cocrystal or salt formers (CCFs), and 4 small glass beads (Ø 3 mm). The grinding program was set for 30 min at a beating frequency of 30 Hz. The resulting ground powders were analysed by X-ray powder diffraction. For the suspected multicomponent crystals of pyruvic acid with caffeine, theophylline, and hypoxanthine, the grinding experiments were repeated using different ratios ranging from 1:1 to 1:4 in order to increase the probability of obtaining cocrystals. The grinding experiments with 1:4 ratios were the only ones that gave conclusive results inciting further study.

### 2.3. Single Crystal Growth

Single crystals were obtained by preparing undersaturated solutions of an equimolar amount (0.3 mmol) of PA and CCFs in a suitable amount of solvent (from 1 to 3 mL). The solutions were left to evaporate slowly (from 3 to 10 days) at room temperature, and single crystals were retrieved. The solvents tested include 2-propanol (IPA), acetone (ACTN), acetonitrile (ACN), chloroform (CHCl$_3$), dichloromethane (DCM), diethyl ether (DEE), ethanol (EtOH), ethyl acetate (EtOAc), methanol (MeOH), tert-butanol (*t*-BuOH), tetrahydrofuran (THF), and water (H$_2$O). Suitable crystals of the 1:1 pyruvic acid-4-nitrobenzamide cocrystal were obtained in ethyl acetate, whereas crystals were harvested from isopropanol

for the 1:1 pyruvic acid-carbamazepine cocrystal, from ethanol for the 1:1 pyruvic acid-isonicotinamide salt, and finally from acetonitrile for the 2:3 pyruvic acid-nicotinamide salt cocrystal.

### 2.4. Congruency Experiments

Congruency experiments were performed through slurry crystallization. Slurry experiments were achieved by suspending equimolar amounts (0.5 mmol) of PA and CCFs in a solvent at 25 °C. The suspensions were stirred at 650 rpm for 3 to 4 days at 25 °C in sealed vials, using a Cooling Thermomixer HLC. After 1h of stirring, each vial was seeded with the corresponding ground powders to assure seeds of potential cocrystals were present. After having reached equilibrium, the powders were filtered, washed, dried, and analysed using PXRD.

### 2.5. Powder X-ray Diffraction (PXRD)

X-ray diffraction measurements were conducted on a Siemens D5000 diffractometer equipped with a Cu cathode ($\lambda$ = 1.5418 Å), operating at 40 kV and 40 mA and using a Bragg Brentano geometry. X-ray patterns were recorded from 5 to 50° in 2θ angles values, with an increment step of 0.02° and an integration time of 2 s (rate of 0.6°/min). Simulated patterns of the known starting compounds were calculated from their single crystal structures with the software Mercury 4.2.0 [27].

### 2.6. Single Crystal Structure Determination

Single crystal X-ray diffraction (SCXRD) analysis was carried out using a MAR345 image plate detector using Mo K$\alpha$ radiation from an Incoatec microfocus source with Montel focusing mirrors. Images were integrated with CrysAlis[PRO], and the implemented absorption correction was applied [28]. Structure solution was carried out by dual space direct methods (SHELXT) [29], and the structure was further refined against $F^2$ using SHELX-2018/3 [30]. Symmetry analysis and validation were checked using PLATON [31]. Pictures were made using the molecular visualization software Mercury 4.2.0 [27].

### 2.7. Thermogravimetric Analysis (TGA)

Thermogravimetric analyses were performed on a Mettler Toledo TGA-STDA 851e, from 25 to 400 °C, at a scanning rate of 10 °C·min$^{-1}$. The solid samples (5 to 10 mg) were placed in aluminium oxide crucibles. The purge gas was nitrogen, with a continuous flow rate of 50 mL·min$^{-1}$. The data were treated with the STARe 12.12 software.

### 2.8. Proton Nuclear Magnetic Resonance ($^1$H NMR)

$^1$H NMR spectra were recorded on a Bruker-300 MHz spectrometer. The powders obtained from the slurry experiments were solubilised in deuterated solvents. $^1$H NMR chemical shifts are reported in parts per million (ppm) relative to the chemical shift of the peak of the deuterated solvent used, chloroform-*d* (CDCl$_3$; 7.26 ppm) or DMSO-d$^6$ ((CD$_3$)$_2$SO; 2.50 ppm). Spectral multiplicities are noted as follows: singlet = s, doublet = d, triplet = t, quartet = q and multiplet = m.

### 2.9. Differential Scanning Calorimetry (DSC)

Differential scanning calorimetry experiments were carried out from 25 to a maximum of 160 °C, with a scanning rate of 5 °K·min$^{-1}$ on a Mettler Toledo DSC 821e. The solid samples (5 to 10 mg) were placed in aluminium pans with perforated lids. The purge gas used was nitrogen, with a continuous flow rate of 50 mL·min$^{-1}$. The data were treated with the STARe 12.12 software.

## 3. Results and Discussion

### 3.1. Cocrystal Screening

Over 70 cocrystal and salt formers (CCFs) (see full list in Supplementary Materials Table S1) were screened with pyruvic acid by neat grinding. These CCFs were chosen using their potential to form non-covalent bonding intermolecular interactions, such as hydrogen bonds, with pyruvic acid, and more precisely their ability to form supramolecular synthon with pyruvic acid's carboxylic acid group. Figure 2 illustrates the supramolecular building blocks that are generally found in pyruvic acid's solid forms, which may be composed of the same functional groups, such as for the homosynthons formed by CCFs' molecules with each other (Figure 2A), or different functional groups, such as for the heterosynthons between pyruvic acid and its CCF (Figure 2B) [32,33]. Therefore, the cocrystal screening list focuses on compounds containing carboxylic acids, amino acids, and amides, with a particular focus on compounds which are GRAS (generally recognized as safe) [34–37].

**(A)** Homosynthon      **(B)** Heterosynthons

**Figure 2.** Examples of supramolecular synthons found in pyruvic acid's solid forms, as discussed in this study. (**A**) Homosynthon formed by two molecules of CCF. (**B**) Heterosynthons formed between one molecule of pyruvic acid and two molecules of CCF. Oxygen atoms in red, nitrogen atoms in blue.

Comparing the PXRD pattern of the ground products with that of the solid parent compound, new solid forms were observed with eight CCFs (Figure 3), with five more samples undergoing amorphization. For the former, four single crystals were sought to fully establish the salt and (salt) cocrystals' formation.

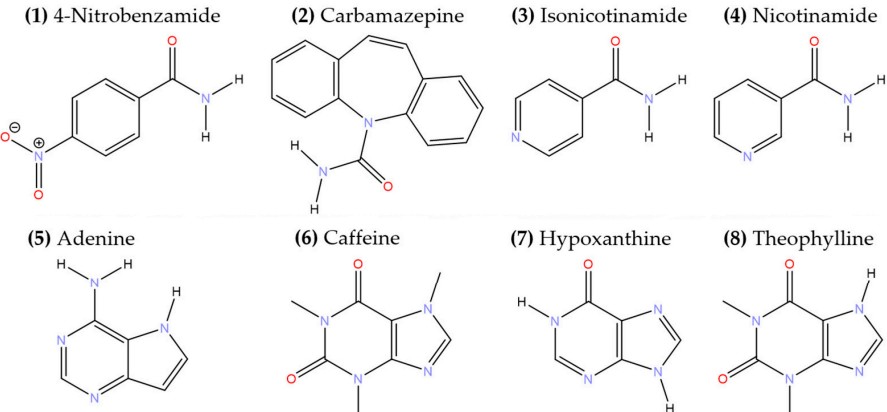

**(1)** 4-Nitrobenzamide  **(2)** Carbamazepine  **(3)** Isonicotinamide  **(4)** Nicotinamide

**(5)** Adenine  **(6)** Caffeine  **(7)** Hypoxanthine  **(8)** Theophylline

**Figure 3.** Chemical structure of the CCFs that led to changes in PXRD patterns upon grinding with pyruvic acid. Oxygen atoms in red, nitrogen atoms in blue.

Here, we confirm the formation of two cocrystals, one salt and one salt cocrystal through single crystal formation. This was the case for 4-nitrobenzamide (**1**), carbamazepine (**2**), isonicotinamide (**3**), and nicotinamide (**4**). Multicomponent crystals such as cocrystal,

salt or salt cocrystal are also suspected for adenine (**5**), caffeine (**6**), hypoxanthine (**7**), and theophylline (**8**), but no single crystal has been obtained so far (see Figures S1–S4 in the SI). The main crystallographic parameters are summarized in Table 1, with a detailed discussion of each below. Full crystallographic data are given in the SI (Supplementary Materials, Tables S2 to S5).

**Table 1.** Main crystallographic parameters of pyruvic acid salt and (salt) cocrystals.

| Solid forms | 1:1 Pyruvic Acid-4-Nitrobenzamide | 1:1 Pyruvic Acid-Carbamazepine | 1:1 Pyruvic Acid-Isonicotinamide | 2:3 Pyruvic Acid-Nicotinamide |
|---|---|---|---|---|
| Abbrev. | PANB | PACBZ | PAINAM | PANAM |
| Structural formula | $C_7H_6N_2O_3$, $C_3H_4O_3$ | $C_{15}H_{12}N_2O$, $C_3H_4O_3$ | $C_6H_7N_2O$, $C_3H_3O_3$ | $2(C_6H_7N_2O)$,$2(C_3H_3O_3)$, $C_3H_4O_3$ |
| FW | 254.19 | 324.33 | 210.19 | 210.19 |
| Crystal system | Primitive Monoclinic | Primitive Monoclinic | Primitive Monoclinic | Primitive Monoclinic |
| Space group | $P2_1/n$ | $P2_1/n$ | $P2_1/n$ | $P2_1/n$ |
| a, b and c (Å) | 5.3463(5), 19.6423(14), 10.8234(7) | 5.3912(6),16.6752(15), 18.2383(18) | 3.8349(6), 33.160(5), 7.6010(11) | 3.82010(12), 33.2921(9), 9.5234(2) |
| α, β and γ (°) | 90, 90.316(7), 90 | 90, 97.324(10), 90 | 90, 97.015(13), 90 | 90, 98.606(3), 90 |
| Cell vol. (Å³) | 1136.59 | 1626.23 | 959.348 | 1197.54 |
| Z | 4 | 4 | 4 | 2 |

### 3.2. Structural and Thermal Characterization of Salt and (Salt) Cocrystals

3.2.1. 1:1 Pyruvic acid-4-Nitrobenzamide Cocrystal (PANB)

Single crystals of the 1:1 Pyruvic Acid-4-Nitrobenzamide cocrystal were obtained by slow evaporation from ethyl acetate. PANB crystallizes in the primitive monoclinic $P2_1/n$ space group and contains four molecules per unit cell. Pyruvic acid and 4-nitrobenzamide molecules bind through four different hydrogen bonds (Figure 4A). This structure is defined as a cocrystal; the acid hydrogen of pyruvic acid may be located in the electron density map and is engaged in a hydrogen bond with the carbonyl group of the amide moiety. Moreover, pyruvic acid's C-O bond distances are not identical (1207(3) and 1308(3) Å). All four hydrogen bonds have $D_1^1(2)$ finite patterns, according to Etter's graph-set notation (Table 2) [32,33].

**Table 2.** Hydrogen bonds in the 1:1 pyruvic acid-4-nitrobenzamide cocrystal.

| Descriptors | Donors | H··· | Acceptors | D-H | H···A | D···A | D-H···A |
|---|---|---|---|---|---|---|---|
| | | | | | **Interatomic Distances (Å)** | | **Angles (°)** |
| $D_1^1(2)$ *a* | N11 | H11A | O22 | 0.86 | 2.14 | 2.982(3) | 164 |
| $D_1^1(2)$ *b* | N11 | H11B | O26 | 0.86 | 2.31 | 3.138(3) | 161 |
| $D_1^1(2)$ *c* | O23 | H23 | O12 | 0.82 | 1.78 | 2.589(3) | 168 |
| $D_1^1(2)$ *a′* | N11 | H11A | O22 | 0.86 | 2.50 | 2.912(3) | 110 |

These interactions result in three coupled intermolecular ring hydrogen-bonding patterns involving the primary amides of 4-nitrobenzamide. A first, an $R_4^4(14)$ ring is formed between the two acceptor carbonyl moieties of pyruvic acid and the two donor moieties of the NH$_2$- group of the primary amide of 4-nitrobenzamide (following the *a* and *b* descriptors through the > *a* < *b* > *a* < *b* path). A second ring motif $R_2^2(8)$ is found between the acceptor carbonyl moiety of the carboxylic acid of pyruvic acid and one donor moiety of the NH$_2$- group of the primary amide of 4-nitrobenzamide, as well as

between the acceptor carbonyl moiety of the primary amide of 4-nitrobenzamide and the donor hydroxyl moiety of the carboxylic acid of pyruvic acid (following the *a* and *c* descriptors through the > *a* < *c* path). A third ring pattern $R_4^4(18)$ is observed between a 4-nitrobenzamide and a pyruvic acid molecule following the *b* and *c* descriptors, through the > *b* > *c* > *b* > *c* path. These hydrogen bonding features lead to an overall 3D « zigzag » network, as shown in Figure 4C. The simulated PXRD pattern matches that obtained during the grinding experiment, showing the same solid form was obtained (Figure 5), albeit there being some parent compound remaining in the ground powder.

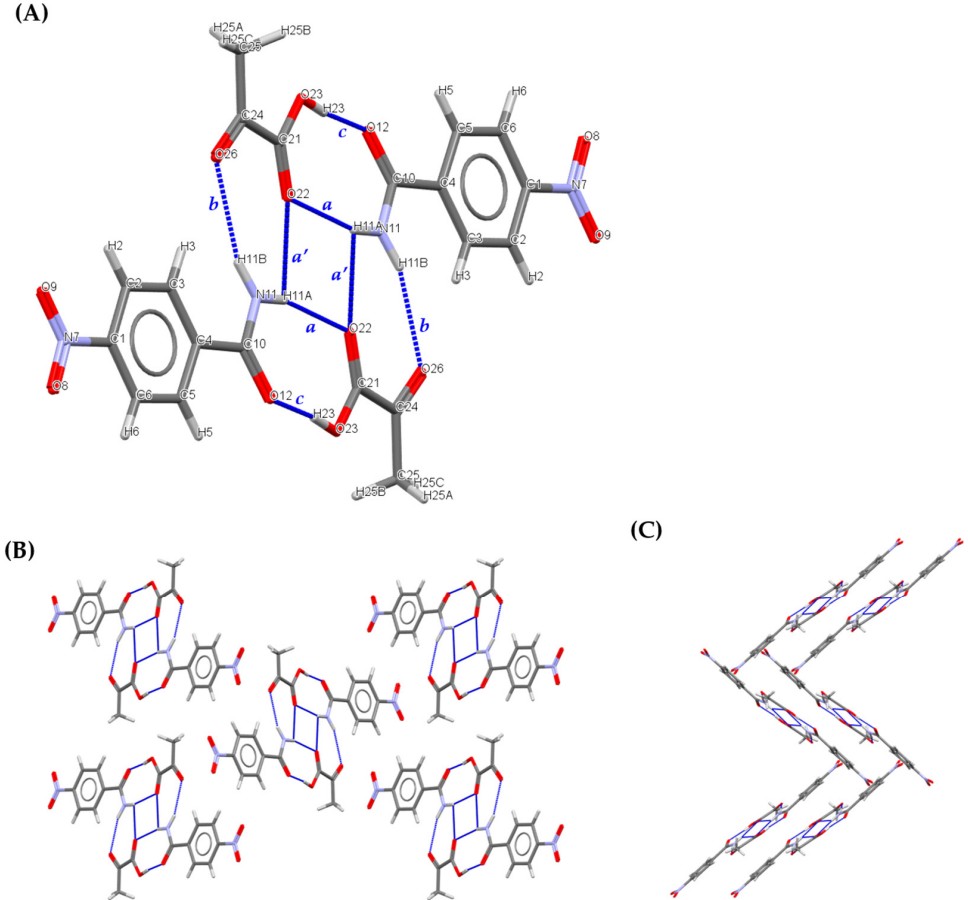

**Figure 4.** Structure of the PANB cocrystal. Hydrogen bonds are highlighted using blue dashed lines, and the atoms implicated in the hydrogen bonds are labelled. (**A**) Hydrogen bond arrangement view along the a-axis. (**B**) Crystal packing view along the a-axis. (**C**) Crystal packing view along the c-axis. Oxygen atoms in red, nitrogen atoms in blue.

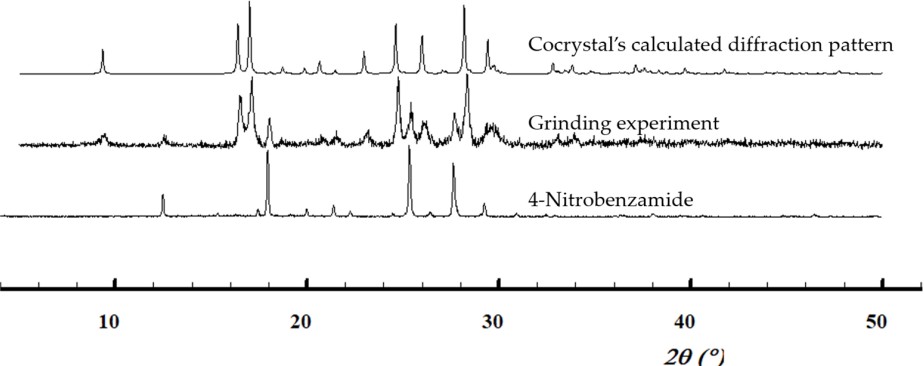

**Figure 5.** Normalized diffraction patterns. Calculated diffraction pattern of pyruvic acid-4-nitrobenzamide cocrystal (PANB) and of 4-nitrobenzamide, and experimental patterns of the ground powder.

A bulk cocrystal material can be obtained from a 1:1 slurry in acetonitrile, ethyl acetate or *tert*-butanol. [1]H NMR analysis (Supplementary Materials, Figure S5) of this powder confirms the 1:1 stoichiometry. Upon heating (Supplementary Materials, Figures S9 and S13), the solid is stable up to 90 °C, at which a first weight loss starts occurring. Full degradation then occurs at 190 °C. The weak signals in the DSC analysis point towards a degradation of the solid form, with no melt occurring.

### 3.2.2. 1:1 Pyruvic Acid-Carbamazepine Cocrystal (PACBZ)

Suitable crystals were grown from isopropanol through evaporative crystallization, allowing the confirmation by SCXRD of a 1:1 cocrystal. PACBZ crystallises in a primitive monoclinic $P2_1/n$ space group, containing four molecules per unit cell. The resolved crystal structure exhibits some static disorder, and we here describe the structure with only one of the orientations of pyruvic acid (Figure 6), as the other orientation is equivalent. Each carbamazepine molecule is shown to share four hydrogen bonds with two different molecules of pyruvic acid, and each molecule of pyruvic acid shares four hydrogen bonds with two different molecules of carbamazepine. The acid hydrogen in the pyruvic acid was located in the electron density maps and is involved in an acid–amide dimer heterosynthon. The C-O bond lengths of pyruvic acid molecules are, moreover, not identical (1.19(2) and 1.31(2) Å). All hydrogen bonds each have $D_1^1(2)$ finite patterns, as detailed in Table 3.

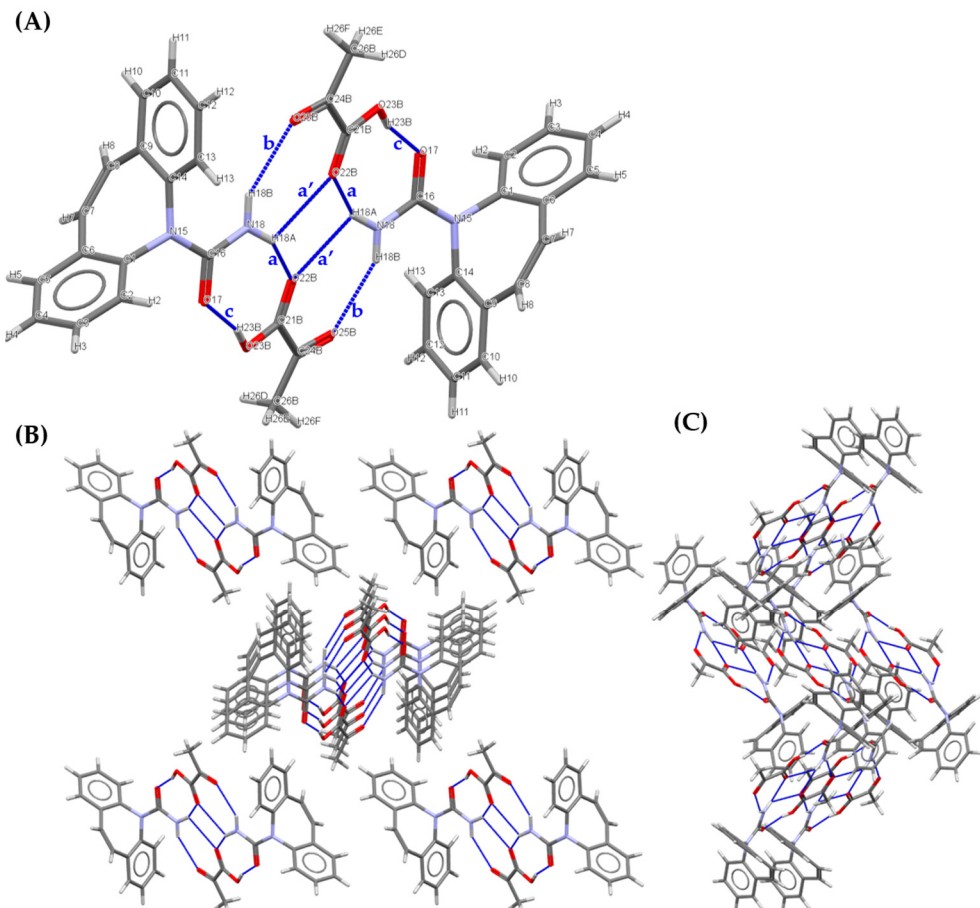

**Figure 6.** Focus on several features of the PACBZ cocrystal. (**A**) Hydrogen bond arrangement view along the a-axis. (**B**) Crystal packing view along the a\*-axis. (**C**) Crystal packing view along the c\*-axis. Nitrogen atoms in blue, oxygen atoms in red.

As for the cocrystal between pyruvic acid and 4-nitrobenzamide, these latter first-level interactions result in three coupled intermolecular ring hydrogen-bond patterns, mainly involving the primary amides of carbamazepine. The first molecule of carbamazepine is

linked by two hydrogen bonds to the same pyruvic acid molecule through an $R_2^2(8)$ ring feature formed on the one hand between an acceptor carbonyl moiety of pyruvic acid and a donor moiety of the NH$_2$- group of the primary amide of carbamazepine (following the *a* descriptor, through the $> a < c$ path), and on the other hand between a donor hydroxyl moiety of pyruvic acid and an acceptor carbonyl moiety of the primary amide of carbamazepine (following the *c* descriptor, through the $> a < c$ path). For the same carbamazepine molecule, another hydrogen bond is formed with a second pyruvic acid molecule, between the acceptor carbonyl moiety of the carboxylic acid of pyruvic acid and the donor NH$_2$- moiety of the primary amide of carbamazepine. This last interaction is involved in a second $R_4^4(14)$ hydrogen-bonding ring pattern, following the path $> a < b > a < b$, using the *a* and *b* descriptors. One last ring pattern $R_4^4(18)$ can also be defined by using the *c* and *b* descriptors, through the $> b > c > b > c$ path. As shown in Figure 6C, a combination of these patterns generates a 3D « zigzag » network. The simulated PXRD pattern matches that obtained during the grinding experiment, showing the same solid form was obtained (Figure 7), albeit there being some parent compound remaining in the ground powder.

**Table 3.** Hydrogen bonds in the 1:1 pyruvic acid-carbamazepine cocrystal.

| Descriptors | Donors | H$\cdots$ | Acceptors | Interatomic Distances (Å) | | | Angles (°) |
| | | | | D-H | H$\cdots$A | D$\cdots$A | D-H$\cdots$A |
|---|---|---|---|---|---|---|---|
| $D_1^1(2)$ *a* | N18 | H18A | O22B | 0.86 | 2.27 | 3.019(17) | 145 |
| $D_1^1(2)$ *b* | N18 | H18B | O25B | 0.86 | 2.32 | 3.101(19) | 152 |
| $D_1^1(2)$ *c* | O23 | H23B | O17 | 0.82 | 1.71 | 2.515(11) | 165 |
| $D_1^1(2)$ *a'* | N18 | H18A | O22B | 0.86 | 2.54 | 2.990(19) | 114 |

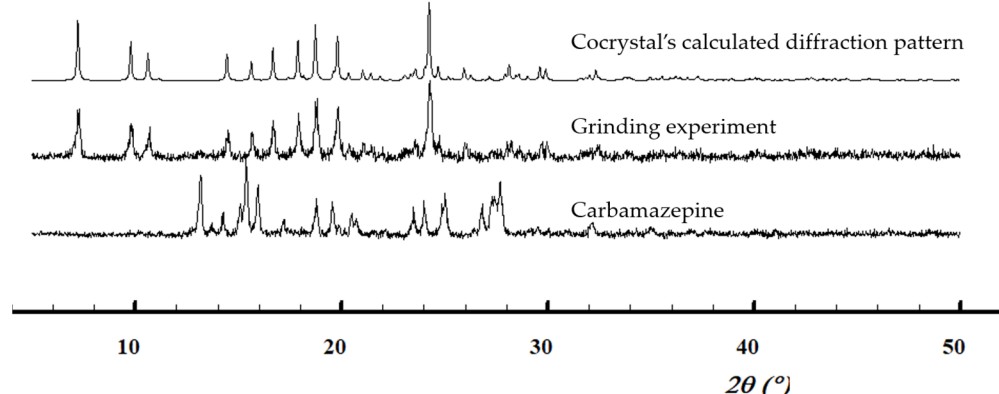

**Figure 7.** Normalised PXRD patterns. Calculated diffraction pattern of pyruvic acid-carbamazepine cocrystal (PACBZ), experimental patterns of the ground powder (1:1 ratio) and of carbamazepine.

The bulk powder can be obtained in solvents such as 2-propanol, acetonitrile, dichloromethane and ethyl acetate, using a ratio of one equivalent of carbamazepine to four equivalents of pyruvic acid. [1]H NMR (Supplementary Materials, Figure S6) confirms the 1:1 stoichiometry. Thermal analysis (Supplementary Materials, Figures S10 and S14) indicates a melting (DSC) occurring at 100 °C, followed by an immediate degradation (TGA).

### 3.2.3. 1:1 Pyruvic Acid-Isonicotinamide Salt (PAINAM)

Single crystals obtained from ethanol show a 1:1 multi-component crystal between pyruvic acid and isonicotinamide. PAINAM crystallises in the primitive monoclinic $P2_1/n$ space group and contains four molecules in the unit cell. Strong hydrogen bonds are usually formed between carboxylic acids, such as the one of pyruvic acid and N-heterocyclic hydrogen-bond acceptors, e.g., the pyridine group of isonicotinamide, and can even form a salt if the proton is completely transferred from the acid to the base. Electron density

analysis has shown that the proton of the carboxylic acid group of pyruvic acid is found on the pyridine nitrogen atom of isonicotinamide, defining this solid form as a salt. The protonated pyridine has, moreover, a C-N-C bond angle of over 120° [38]. Nevertheless, pyruvic acid's C-O bond distances are not identical (1214(6) and 1292(6) Å), which means PAINAM can be found in the grey area of the salt–cocrystal continuum, lying between a true salt and a true cocrystal.

Isonicotinamide molecules are linked together via two hydrogen bonds through their primary amide groups by an $R_2^2(8)$ ring feature, following the *b* descriptor, as detailed in Table 4, and illustrated in Figure 8, more precisely through the donor -NH$_2$ moiety of a primary amide and the acceptor carbonyl moiety of another primary amide. Every isonicotinamide molecule is also shown to share two hydrogen bonds with two different pyruvic acid molecules, and these hydrogen bonds each have a $D_1^1(2)$ finite pattern following the *a* or the *c* descriptors. The first hydrogen bond is formed between the donor NH- moiety of a pyridine ring (protonated isonicotinamide) and the acceptor carbonyl moiety of a carboxylic acid (pyruvate), following the descriptor *a*. The second hydrogen bond links the donor NH$_2$- moiety of an amide group (isonicotinamide) and the acceptor carbonyl moiety of a carboxylic acid (pyruvic acid) following the *c* descriptor.

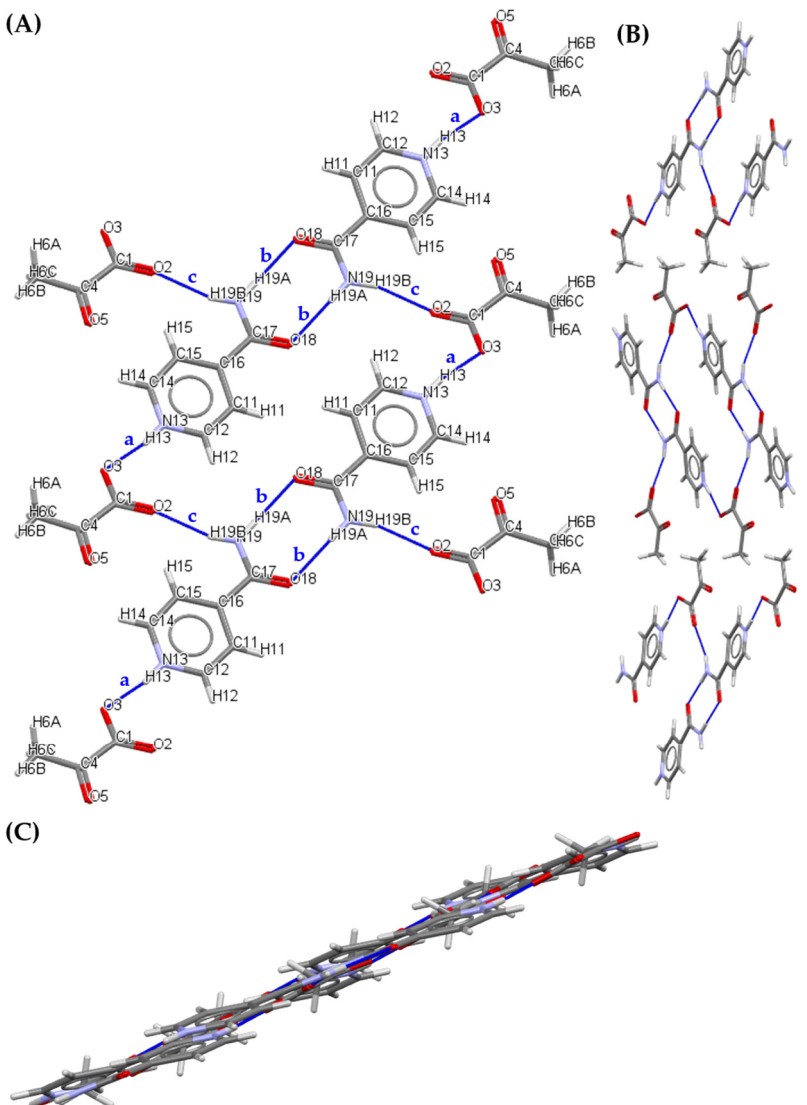

**Figure 8.** PAINAM structural features. (**A**) Hydrogen bond arrangement view along the a-axis. (**B**) Crystal packing view along the c*-axis (view down reciprocal cell axis c*). (**C**) Crystal packing view along the b-axis. Oxygen atoms in red, nitrogen atoms in blue.

**Table 4.** Hydrogen bonds in the 1:1 pyruvic acid-isonicotinamide salt.

| Descriptors | Donors | H··· | Acceptors | Interatomic Distances (Å) | | | Angles (°) |
| --- | --- | --- | --- | --- | --- | --- | --- |
| | | | | D-H | H···A | D···A | D-H···A |
| $D_1^1(2)\,a$ | N13 | H13 | O3 | 0.86 | 1.72 | 2.566(5) | 169 |
| $D_1^1(2)\,c$ | N19 | H19B | O2 | 0.86 | 2.07 | 2.894(5) | 159 |
| $R_2^2(8) > b > b$ | N19 | H19A | O18 | 0.86 | 2.07 | 2.929(5) | 179 |

These interactions result in a finite pattern $D_3^3(17)$ following the *a* and *b* descriptors ($< a > b > a$). Another path is also described through a $D_3^3(9)$ finite pattern, following the *c* and *b* descriptors ($< c > b > c$). The last Etter graph is identified between pyruvic acid and isonicotinamide molecules through $C_2^2(11)$ intermolecular hydrogen bonds, which link one pyruvate molecule to two isonicotinamide molecules, following the *a* and *c* descriptors ($> a < c$). Overall, these bonding patterns generate a 3D layered structure, as shown in Figure 8C. The simulated PXRD pattern matches that obtained during the grinding experiment, showing the same solid form was obtained (Figure 9), albeit some parent compound remains in the ground powder.

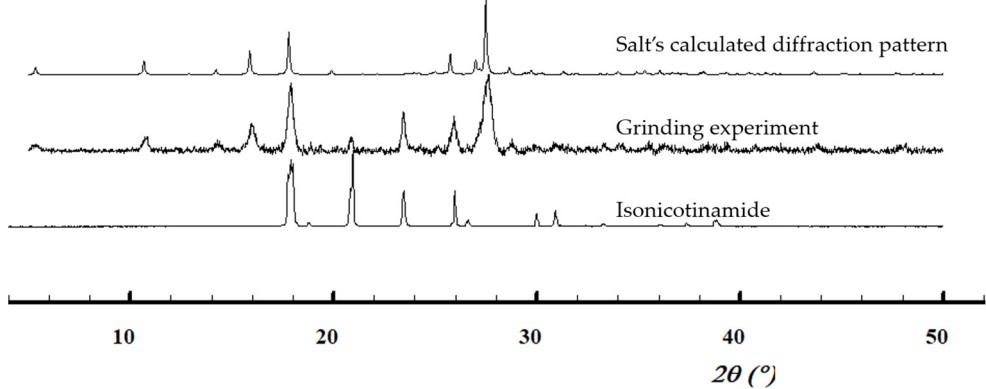

**Figure 9.** PXRD calculated diffraction pattern of pyruvic acid-isonicotinamide salt (PAINAM) and of isonicotinamide, and experimental patterns of the ground powder.

The bulk powder was obtained from several solvents: 2-propanol, acetonitrile, dichloromethane, ethanol, ethyl acetate, methanol, *tert*-butanol, and water. [1]H NMR analysis confirms a 1:1 stoichiometry, validating the SCXRD results (Supplementary Materials, Figure S7). Thermal analysis shows a melt (Supplementary Materials, Figure S15) followed by immediate degradation (Supplementary Materials, Figure S11) at about 120 °C.

### 3.2.4. 2:3 Pyruvic Acid-Nicotinamide Salt Cocrystal (PANAM)

Single crystals of the 2:3 pyruvic acid-nicotinamide were obtained from acetonitrile. PANAM crystallises in the monoclinic $P2_1/n$ space group, with the asymmetric unit containing one fully occupied nicotinamide molecule and a fully occupied pyruvic acid molecule, as well as a half-occupied pyruvic acid molecule found disordered on an inversion centre. The following description relates to either of the equivalent orientation of the pyruvic acid molecule, as shown in Figure 10. In this structure, two types of pyruvic acid can be found: deprotonated or protonated. The former links to the pyridine group of nicotinamide, with C-O bonds of similar lengths (1.219(3) and 1.251(2) Å), and refinement of the electron density, showing the proton to be closer to the pyridine nitrogen atom. Furthermore, the protonated pyridine has a C-N-C bond angle of 121.82°. In contrast, the pyruvic acid molecule not bound to nicotinamide shows no proton transfer, with C-O and C=O bond lengths of, respectively, 1.202 and 1.294 Å. PANAM can therefore be defined as a salt cocrystal, being a cocrystal between pyruvic acid and the pyruvic acid-nicotinamide salt.

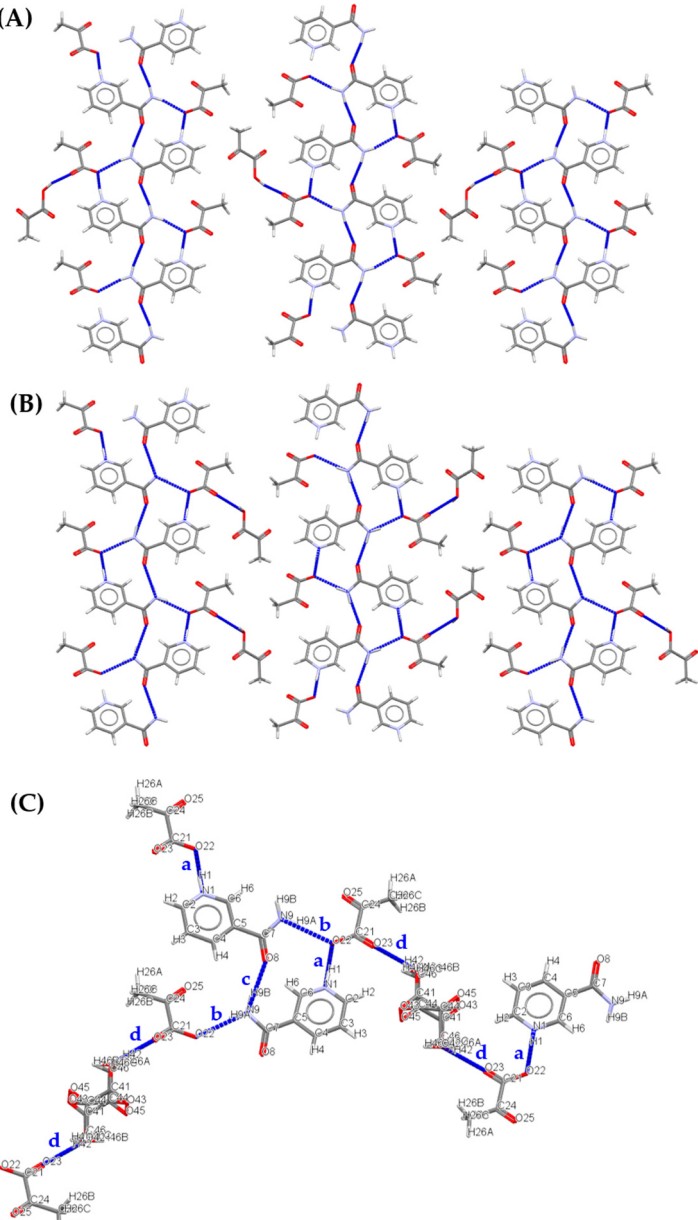

**Figure 10.** Focus on structural features of the salt cocrystal PANAM. (**A**) Crystal packing view along the a-axis for the first orientation. (**B**) Crystal packing view along the a-axis for the second orientation. (**C**) Hydrogen bond arrangement view along the a-axis (with disorder). Oxygen atoms in red, nitrogen atoms in blue.

In the structure, protonated nicotinamide molecules form chains of molecules that are surrounded by pyruvic acid/pyruvate molecules. As described in Table 5, each protonated nicotinamide molecule is linked to two other protonated nicotinamide molecules via hydrogen bonds through their primary amide groups, more precisely through the donor -$NH_2$ moiety of a primary amide and the acceptor carbonyl moiety of another primary amide, following the *c* descriptor, in a $C_1^1(4)$ chain pattern. Each nicotinamide molecule is also linked to two pyruvate molecules, also in $D_1^1(2)$ finite patterns. The first hydrogen bond is found between the donor NH- moiety of a pyridine ring of a protonated nicotinamide molecule and the acceptor carbonyl moiety of a carboxylate of a pyruvate molecule, following the descriptor *a*. The second hydrogen bond is formed between the donor $NH_2$- moiety of an amide group of a protonated nicotinamide molecule and the acceptor carbonyl moiety of a pyruvate molecule, following the *b* descriptor. Pyruvate molecules have the role of linkers

between two molecules of protonated nicotinamide, but are also bound to a pyruvic acid molecule through a $D_1^1(2)$ hydrogen bond finite pattern, following the *d* descriptor.

**Table 5.** Hydrogen bonds in the 2:3 pyruvic acid-nicotinamide salt cocrystal.

| Descriptors | Donors | H··· | Acceptors | Interatomic Distances (Å) | | | Angles (°) |
| | | | | D-H | H···A | D···A | D-H···A |
|---|---|---|---|---|---|---|---|
| $D_1^1(2)$ *a* | N18 | H18A | O22B | 0.86 | 2.27 | 3.019(17) | 145 |
| $D_1^1(2)$ *b* | N18 | H18B | O25B | 0.86 | 2.32 | 3.101(19) | 152 |
| $D_1^1(2)$ *c* | O23 | H23B | O17 | 0.82 | 1.71 | 2.515(11) | 165 |
| $D_1^1(2)$ *a′* | N18 | H18A | O22B | 0.86 | 2.54 | 2.990(19) | 114 |

These bonds lead to $C_2^1(8)$ intermolecular patterns, which link one pyruvate molecule to two nicotinamide molecules, following > *a* < *b* path or a $D_3^3(15)$ finite pattern, which binds a nicotinamide molecule to another nicotinamide molecule and a pyruvate molecule, following the < *a* > *c* > *a* path. Other paths are also found, such as through the $D_2^2(5)$ finite pattern following the *a* and *d* descriptors (> *a* < *d*), as well as through the $D_3^3(9)$ finite pattern following the *b* and *c* descriptors (< *b* > *c* > *b*). The ultimate Etter graph is identified around a pyruvate linker molecule following an *a* > *b* > *d* path. The chain of protonated nicotinamide molecules forms, simply overlapping layers of molecules. The simulated PXRD pattern (Figure 11) shows the same form was found during the grinding experiments), albeit some parent compound remains in the ground powder.

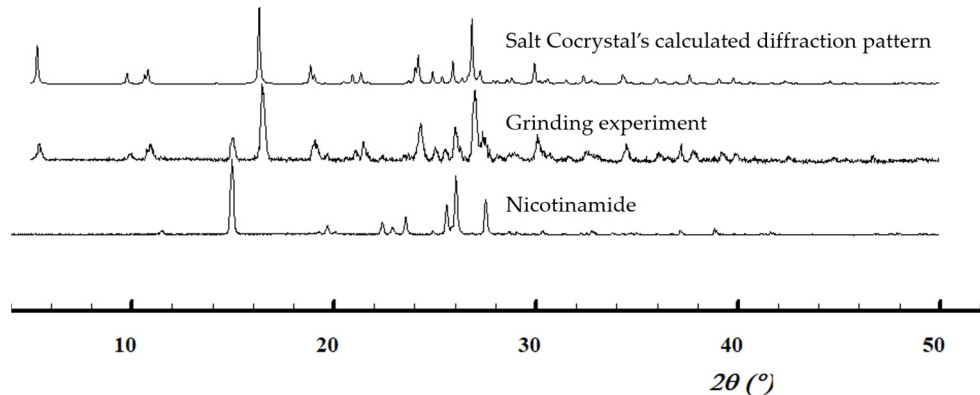

**Figure 11.** Normalised PXRD patterns. Calculated diffraction patterns of pyruvic acid-nicotinamide salt cocrystal (PANAM) and nicotinamide, and the experimental pattern of the ground powder.

Bulk 2:3 pyruvic acid-nicotinamide was obtained from 2-propanol and ethyl acetate. [1]H NMR analysis validates a 2:3 stoichiometry (Supplementary Materials, Figure S8). PANAM shows a melting temperature of 73.4 °C (Supplementary Materials, Figure S16), with degradation of the melt occurring at around 90 °C (Supplementary Materials, Figure S12).

Table 6 summarizes the four cases discussed above, highlighting the potential of cocrystallisation and salt formation to enhance the thermal stability of pyruvic acid. An example of TGA and DSC combination can be found Supplementary Materials, Figure S17. In general, the solid powders remained thermally stable up to temperatures between 70 and 120 °C, depending on the nature of the cocrystal or salt former with a melting/degradation point much higher than the starting material (11.8 °C). These results prove that crystal engineering is a useful tool to raise APIs melting points and thermally stabilize liquid compounds into different crystalline forms, which are a stable and convenient way to transport and store a molecule. Undergoing growing demand, the solidification of pyruvic acid could in the future facilitate the handling of this liquid compound at room temperature. This work could pave the way for the establishment of a general method for solidifying

low-melting compounds. The latter would be very useful industrially, since drugs are usually favoured in a solid dosage form.

**Table 6.** Solid forms identified for pyruvic acid, together with the melting temperatures or degradation temperatures ranges.

| Cocrystal/Salt Former | Ratio | Formed Solid | M.P./Degradation Range (°C) |
|---|---|---|---|
| 4-Nitrobenzamide | 1:1 | Cocrystal | 90–100 |
| Carbamazepine | 1:1 | Cocrystal | 100 |
| Isonicotinamide | 1:1 | Salt | 120–130 |
| Nicotinamide | 2:3 | Salt cocrystal | 73 |

## 4. Conclusions

In this study, we highlight the potential of crystal engineering to stabilize liquid compounds such as pyruvic acid by playing on the nature of the cocrystal and salt formers. Pyruvic acid cocrystal and salt screening led to the identification of eight novel solid forms. Among these, four systems (with 4-nitrobenzamide, carbamazepine, isonicotinamide and nicotinamide) were confirmed by single crystal analysis, with four other solid forms (with adenine, caffeine, hypoxanthine and theophylline) also likely forming multicomponent crystals. The thermic stability of the confirmed solid forms of pyruvic acid was evaluated, showing the solid forms to be stable up to temperatures between 70 and 120°C. Such solid forms, with GRAS compounds, facilitate the handling of pyruvic acid in and potentially widen its field of application to the pharmaceutical, cosmetic, food, and chemical industries.

**Supplementary Materials:** The following supporting information can be downloaded at: https://www.mdpi.com/article/10.3390/cryst13050808/s1. CCDC 2258062-2258065 contains the supplementary crystallographic data for this paper. These data can be obtained free of charge from The Cambridge Crystallographic Data Centre via www.ccdc.cam.ac.uk/structures, accessed on 8 May 2023.

**Author Contributions:** Conceptualization, T.L., D.P.D. and P.L; methodology, T.L. and C.C.G.; software, K.R.; validation, T.L. and K.R.; formal analysis, C.C.G.; investigation, C.C.G.; resources, T.L., D.P.D. and P.L.; data curation, C.C.G.; writing—original draft preparation, C.C.G.; writing—review and editing, T.L., D.P.D., K.R. and C.C.G.; supervision, T.L.; project administration, T.L.; funding acquisition, T.L., P.L. and D.P.D. All authors have read and agreed to the published version of the manuscript.

**Funding:** This research was funded by FNRS (J.0168.22) and Actions de Recherche Concertée ARC, grant number 20-25/108.

**Data Availability Statement:** The data presented in this study are available in Supporting Information.

**Conflicts of Interest:** The authors declare no conflict of interest.

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
