# Peer review of "From Liquid to Solid: Cocrystallization as an Engineering Tool for the Solidification of Pyruvic Acid"

_crystals, doi:10.3390/cryst13050808_

Round 1
Reviewer 1 Report
This manuscript describes the synthesis and single crystal structures of four cocrystals of pyruvic acid with other materials. This strategy of co-crystallization is a common process, both synthetically and naturally. Notably, this is the same process as the formation of crystalline solvates. The solvate, or co-crystal, will always have a lower melting temperature than that of the pure phase of the high melting component, but will often exhibit a higher melting point that that of the low-melting member of the pair. The authors have effectively used this strategy to prepare solid pyruvic acid materials.
Overall, this manuscript is appropriate for publication in Crystals. The following suggestions would improve communication.
The language of homo and hetero synthons is introduced in Fig. 2 but then is not used in the manuscript. In fact Fig. 2 does not include pyruvic acid in any of the synthon schematics…raising question as to whether they should identified as “synthons of pyruvic acid’s solid forms.” Such schematics would be more helpful if they demonstrated how pyruvic acid can interact with different hydrogen bonding functionalities. Further, this would be an effective place to introduce the Etter graphical notation, which is used to describe the crystal structures, but never explained.
P. 4, L. 149 it is not clear what the authors mean by multicomponent crystals. By one definition co-crystals are multicomponent, though thermodynamically the form a single component compound. Do they mean polycrystalline samples for which single crystals could not be grown? Or do they mean that multiple phases are present?
All of the crystal structure figures are hard to read and identify components. I would recommend modifying the color scheme to emphasize the pyruvic acid component of the crystal structure. The blue dotted lines largely print as solid lines. Greater separation between dots or dashes is needed to differentiate them from non-H-bonds. It would be useful to use a more contrasting blue color for the N, as with the current color scheme it is difficult to differentiate the N and C. The font size for the labeling scheme should also be increased since it is almost unreadable at the current scale.
In Figs. 5, 7 and 11, the diffraction pattern should be identified as a calculated diffraction pattern, not a spectrum. Diffraction patterns are not spectra.
Table 6 is the first place there is indication of the TGA and DSC data identified in the experimental section. It would be useful to show a figure, at least in supplementary materials that compares the TGA and DSC of these materials rather than just listing a numerical M.P./degradation range value in a table.
Related to the above, there is some confusion with the use of the term stable p. 11 L. 342. I presume they are referring to thermal stability relative to a TGA trace. However, one of the stability issues noted in the introduction was the hydroscopic behavior of pyruvic acid. Are these cocrystals more stable with respect to hydration? Or just with respect to thermal decomposition?
Author Response
We thank the reviewer and answered positively to most suggestions:
- "The language of homo and hetero synthons is introduced in Fig. 2 but then is not used in the manuscript. In fact Fig. 2 does not include pyruvic acid in any of the synthon schematics…raising question as to whether they should identified as “synthons of pyruvic acid’s solid forms.” Such schematics would be more helpful if they demonstrated how pyruvic acid can interact with different hydrogen bonding functionalities. Further, this would be an effective place to introduce the Etter graphical notation, which is used to describe the crystal structures, but never explained." The text has been clarified and an image added to explain this point.
- "P. 4, L. 149 it is not clear what the authors mean by multicomponent crystals. By one definition co-crystals are multicomponent, though thermodynamically the form a single component compound. Do they mean polycrystalline samples for which single crystals could not be grown? Or do they mean that multiple phases are present? We specifially clarified what we mean by multi-component systems.
- All of the crystal structure figures are hard to read and identify components. I would recommend modifying the color scheme to emphasize the pyruvic acid component of the crystal structure. The blue dotted lines largely print as solid lines. Greater separation between dots or dashes is needed to differentiate them from non-H-bonds. It would be useful to use a more contrasting blue color for the N, as with the current color scheme it is difficult to differentiate the N and C. The font size for the labeling scheme should also be increased since it is almost unreadable at the current scale As this is a recommendation, we prefer keeping our current labeling and representation scheme.
- "In Figs. 5, 7 and 11, the diffraction pattern should be identified as a calculated diffraction pattern, not a spectrum. Diffraction patterns are not spectra." Changed.
- Table 6 is the first place there is an indication of the TGA and DSC data identified in the experimental section. It would be useful to show a figure, at least in supplementary materials that compare the TGA and DSC of these materials rather than just listing a numerical M.P./degradation range value in a table. this has been done. We do mention that the analyses were taken with different devices, and using therefore different conditions (flow, amount of material). A direct comparison is difficult, but we did this for the supporting information as suggested by the reviewer.
-
"Related to the above, there is some confusion with the use of the term stable p. 11 L. 342. I presume they are referring to thermal stability relative to a TGA trace. However, one of the stability issues noted in the introduction was the hydroscopic behavior of pyruvic acid. Are these cocrystals more stable with respect to hydration? Or just with respect to thermal decomposition?"
We added the word 'thermal' to clarify that we indeed talk about thermal stability.
Reviewer 2 Report
The paper reports the screening by neat grinding of pyruvic acid with 73 coformers. Eight crystalline forms were identified, being the crystalline structures of 2 cocrystals, 1 salt and 1 cocrystal salt solved. Five amorphous forms were also identified. This work is very interesting as multicomponent solid forms were obtained in which one of the co-formers is liquid at room temperature, which may increase the applicability field of pyruvic acid. The solids whose crystalline structures were solved were also characterized by thermogravimetric analysis, differential scanning calorimetry and proton nuclear magnetic resonance.
The paper is very well written, with clear structure and careful explanations throughout, enabling others to replicate these techniques if desired. The quality of experimental data is convincing and the conclusions appear to be reliable. I have just two comments which the authors may wish to address:
1) On page 2, line 73, the author’s state that grinding experiments were repeated using 1:4 ratios with caffeine, theophylline and hypoxanthine, but it is not explained why they ground with this molar ratio nor what conclusions were reached.
2) In the description of the thermal behavior, concerning the systems Isonicotinamide-Pyruvic acid and Nicotinamide-Pyruvic acid, by TGA and DSC there is a mistake. In page 10 line 285, melt is observed in Figure S15 and degradation in Figure S11, not the opposite as stated in the text. On page 11, lines 334 and 335, Figure S12 should be Figure S16 and vice versa.
Author Response
Changes and mistakes noted by the reviewer have been adapted in the new version.